# Perfect Match: A Simple Method for Learning Representations For Counterfactual Inference With Neural Networks

## Abstract

Learning representations for counterfactual inference from observational data is of high practical relevance for many domains, such as healthcare, public policy and economics. Counterfactual inference enables one to answer "What if...?" questions, such as "What would be the outcome if we gave this patient treatment $t_1$?". However, current methods for training neural networks for counterfactual inference on observational data are either overly complex, limited to settings with only two available treatment options, or both. Here, we present Perfect Match (PM), a method for training neural networks for counterfactual inference that is easy to implement, compatible with any architecture, does not add computational complexity or hyperparameters, and extends to any number of treatments. PM is based on the idea of augmenting samples within a minibatch with their propensity-matched nearest neighbours. Our experiments demonstrate that PM outperforms a number of more complex state-of-the-art methods in inferring counterfactual outcomes across several real-world and semi-synthetic datasets.

## 1 Introduction

Estimating individual treatment effects[1] (ITE) from observational data is an important problem in many domains. In medicine, for example, we would be interested in using data of people that have been treated in the past to predict what medications would lead to better outcomes for new patients (Shalit et al. (2017)). Similarly, in economics, we would for example want to determine how effective job programs would be based on results of past job training programs (LaLonde (1986)).

ITE estimation from observational data is difficult for two reasons: Firstly, we never observe all potential outcomes. If a patient is given a treatment to treat her symptoms, we never observe what *would have happened if* the patient was prescribed a potential alternative treatment in the same situation. Secondly, the assignment of cases to treatments is typically biased such that cases for which a given treatment is more effective are more likely to have received that treatment. The distribution of samples may therefore differ significantly between the treated group and the overall population. A supervised model naïvely trained to minimise the factual error would overfit to the properties of the treated group, and thus not generalise well to the entire population.

To address these problems, we introduce Perfect Match (PM), a simple method for training neural networks for counterfactual inference that extends to any number of treatment options. PM effectively controls for biased assignment of treatments in observational data by augmenting every sample within a minibatch with its closest matches by propensity score from the other treatment options. PM is easy to use with existing neural network architectures, simple to implement, and does not add any hyperparameters or computational complexity. We perform experiments that demonstrate that PM outperforms a number of more complex state-of-the-art methods in inferring counterfactual outcomes on several real-world and semi-synthetic datasets. Our experiments also show that PM is more robust to a high level of treatment assignment bias than existing methods. We believe that our simple and effective method for ITE estimation that extends to any number of treatment options could potentially make counterfactual inference more accessible, particularly for medical applications, where multiple available treatment options are the norm.

---

[1]The ITE is sometimes referred to as Conditional Average Treatment Effect (CATE).

**Contributions.** This work contains the following contributions:

- We introduce Perfect Match, a simple methodology for learning neural representations for counterfactual inference based on propensity score matching within minibatches.
- We perform extensive experiments on semi-synthetic and real-world data in both the binary and multiple treatment setting. The experiments show that PM outperforms a number of more complex state-of-the-art methods in inferring counterfactual outcomes.

## 2 RELATED WORK

There are five main categories of approaches to learning to estimate ITEs:

**Matching-based Methods.** Matching methods estimate the counterfactual outcome of a sample $X$ with respect to treatment $t$ using the factual outcomes of its nearest neighbours having received $t$, with respect to a metric space. k-Nearest-Neighbour (kNN) (Ho et al. (2007)) operates in the potentially high-dimensional covariate space, and therefore might suffer from the curse of dimensionality. Propensity Score Matching (PSM) (Rosenbaum & Rubin (1983)) matches on the scalar probability $p(t|X)$ of $t$ given the covariates $X$. PSM may require under- or oversampling the original data.

**Adjusted Regression Methods.** Adjusted regression models apply regression models with both treatment and covariates as input. The simplest case is Ordinary Least Squares, which can either be used for building one model, with the treatment as an input feature, or multiple separate models, one for each treatment (Kallus (2017)). More complex regression models, such as Treatment-Agnostic Representation Networks (TARNET) (Shalit et al. (2017)) may be used to capture non-linear relationships. Methods that make use of propensity scores, and a model of the outcomes, such as Propensity Dropout (PD) (Alaa et al. (2017)), are referred to as doubly robust (Funk et al. (2011)).

**Tree-based Methods.** Tree-based methods train many weak learners to build expressive ensemble models. Examples of tree-based methods are Bayesian Additive Regression Trees (BART) (Chipman et al. (2010); Chipman & McCulloch (2016)) and Causal Forests (CF) (Wager & Athey (2017)).

**Representation-balancing Methods.** Representation-balancing methods seek to learn a high-level representation for which the covariate distributions are balanced across treatment groups. Balancing Neural Networks (Johansson et al. (2016)) attempt to find such representations by minimising the discrepancy distance (Mansour et al. (2009)) between treatment groups. Counterfactual Regression Networks (CFRNET, Shalit et al. (2017)) use different metrics such as the Wasserstein distance.

**Distribution-modeling Methods.** Generative Adversarial Nets for inference of Individualised Treatment Effects (GANITE) (Yoon et al. (2018)) address ITE estimation using counterfactual and ITE generators. GANITE uses a complex architecture with many hyperparameters and submodels that may be difficult to implement and optimise. Causal Effect Variational Autoencoders (CEVAEs) (Louizos et al. (2017)) use a latent variable modeling approach that is robust to hidden confounding, but does not address biased treatment assignment. Causal Multi-task Gaussian Processes (CMGP) (Alaa & van der Schaar (2017)) apply a multi-task Gaussian Process to ITE estimation. The optimisation of CMGPs involves a matrix inversion of $O(n^3)$ complexity that limits their scalability.

In contrast to existing methods, PM is a simple doubly robust method built on propensity score matching that can be used to train expressive non-linear neural network models for ITE estimation from observational data in settings with any number of treatments (Table 1). While the underlying idea behind PM is simple and effective, it has, to the best of our knowledge, not yet been explored.

Table 1: Comparison of several representative state-of-the-art methods for counterfactual inference showing whether they were designed for $> 2$ treatments and their complexity of implementation.

|  | PM | CF | BART | BNN | CFRNET | PD | CEVAE | CMGP | GANITE |
|---|---|---|---|---|---|---|---|---|---|
| $> 2$ Treatments | ✓ | ✓ | ✓ | ✗ | ✗ | ✗ | ✗ | ✗ | ✓ |
| Complexity | Low | Low | Low | Low | Medium | Medium | High | High | High |

## 3 METHODOLOGY

**Problem Setting.** We consider a setting in which we are given $N$ observed samples $X$, where each sample consists of $p$ covariates $x_i$ with $i \in [0, p)$. For each sample, the potential outcomes are represented as a vector $Y$ with $k$ entries $y_j$ where each entry corresponds to the outcome when applying one treatment $t_j$ out of the set of $k$ available treatments $T = \{t_0, ..., t_{k-1}\}$ with $j \in [0, k)$. As training data, we receive observed samples $X$ and their factual outcomes $y_j$ when applying one treatment $t_j$. The set of available treatments can contain two or more treatment options. We refer to the special case of two available treatment options as the binary treatment setting. Given the observational training data with factual outcomes, we wish to train a predictive model that is able to accurately produce a predicted potential outcomes vector $\hat{Y}$ with $k$ entries $\hat{y}_j$. In literature, this setting is known as the Rubin-Neyman potential outcomes framework (Rubin (2005)).

**Precision in Estimation of Heterogenous Effect (PEHE).** The primary metric that we optimise for when training models to estimate ITE is the PEHE (Hill (2011)). In the binary setting, the PEHE measures the ability of a predictive model to estimate the difference in effect between two treatments $t_0$ and $t_1$ for samples $X$. To compute the PEHE, we measure the mean squared error between the true difference in effect $y_1(n) - y_0(n)$, drawn from the noiseless underlying outcome distributions $\mu_1$ and $\mu_0$, and the predicted difference in effect $\hat{y}_1(n) - \hat{y}_0(n)$ indexed by $n$ over $N$ samples:

$$\epsilon_{\text{PEHE}} = \frac{1}{N} \sum_{n=0}^{N} \left( \mathbb{E}_{y_j(n) \sim \mu_j(n)} [y_1(n) - y_0(n)] - [\hat{y}_1(n) - \hat{y}_0(n)] \right)^2 \qquad (1)$$

When the underlying noiseless distributions $\mu_j$ are not known, the true difference in effect $y_1(n) - y_0(n)$ can be estimated using the noisy ground truth outcomes $y_i$ (Appendix A). As a secondary metric, we consider the error $\epsilon_{\text{ATE}}$ in estimating the average treatment effect (ATE) (Hill (2011)). The ATE measures the average difference in effect across the whole population (Appendix B). The ATE is not as important as PEHE for models optimised for ITE estimation, but can be a useful indicator of how well an ITE estimator performs at comparing two treatments across the entire population. We can neither calculate $\epsilon_{\text{PEHE}}$ nor $\epsilon_{\text{ATE}}$ without knowing the outcome generating process.

**Multiple Treatments.** Both $\epsilon_{\text{PEHE}}$ and $\epsilon_{\text{ATE}}$ can be trivially extended to multiple treatments by considering the average PEHE and ATE between every possible pair of treatments. Note that we lose the information about the precision in estimating ITE between specific pairs of treatments by averaging over all $\binom{k}{2}$ pairs. However, one can inspect the pair-wise PEHE to get the whole picture.

$$\hat{\epsilon}_{\text{mPEHE}} = \frac{1}{\binom{k}{2}} \sum_{i=0}^{k-1} \sum_{j=0}^{i-1} \hat{\epsilon}_{\text{PEHE},i,j} \qquad (2) \qquad \hat{\epsilon}_{\text{mATE}} = \frac{1}{\binom{k}{2}} \sum_{i=0}^{k-1} \sum_{j=0}^{i-1} \hat{\epsilon}_{\text{ATE},i,j} \qquad (3)$$

**Perfect Match (PM).** We consider fully differentiable neural network models optimised via mini-batch stochastic gradient descent to predict potential outcomes $\hat{Y}$ given a sample $X$. To address the treatment selection bias inherent in observational data, we propose the use of a training methodology based on propensity score matching. A propensity score is the conditional probability $p(t|X)$ of a given sample $X$ receiving a specific treatment $t$ (Rosenbaum & Rubin (1983); Ho et al. (2007)). The propensity score is a *balancing score*, meaning that if the distribution of propensity scores across treatment groups is the same, then the distribution of their covariates $x_i$ is also guaranteed to be the same across treatment groups (Ho et al. (2007)). By matching samples with their nearest neighbours by propensity score we can, in the optimal case, balance the covariates $x_i$. This breaks the dependence of treatment assignment on $X$ if all relevant covariates are observed, and therefore effectively removes any potential treatment assignment bias (Ho et al. (2007)). However, in practice, we do not have access to the true treatment assignment probability $p(t|X)$. We therefore have to estimate the propensity score $p(t|X)$ by training a predictive model to predict the likelihood of receiving a treatment $t$ for a given sample $X$. Note that propensity scores are not the only balancing score (Ho et al. (2007)). For example, $X$ itself is another balancing score. However, matching on the one-dimensional propensity score is preferable because it avoids the curse of dimensionality that would be associated with matching on the potentially high-dimensional $X$ directly. In PM, we match every sample within a minibatch with its nearest neighbours by propensity score from all other treatments (Algorithm 1, more details in Appendix C), taken from the training set. Every minibatch the model

is trained on therefore contains the same number of samples for each treatment group, and the covariates $x_i$ of each treatment group are approximately balanced on average. The intuition behind matching on the minibatch level, rather than the dataset level (Ho et al. (2011)), is that it reduces the variance during training which in turn leads to better expected performance for counterfactual inference (Appendix D). In this sense, PM can be seen as a minibatch sampling strategy (Csiba & Richtárik (2018)) specifically designed to improve learning for counterfactual inference.

---

**Algorithm 1** Batch Augmentation using Perfect Match (PM). After augmentation, each batch contains an equal number of samples from each treatment group and the covariates $x_i$ across all treatment groups are approximately balanced.

---

**Input:** Batch of $B$ random samples $X_{\text{batch}}$ with assigned treatments $t$, training set $X_{\text{train}}$ of $N$ samples, number of treatment options $k$, propensity score estimator $E_{\text{PS}}$ to calculate the probability $p(t|X)$ of treatment assigned given a sample $X$
**Output:** Batch $X_{\text{out}}$ consisting of $B \times k$ matched samples
  1: **procedure** PERFECT_MATCH:
  2:      $X_{\text{out}} \leftarrow$ Empty
  3:    **for** sample $X$ with treatment $t$ in $X_{\text{batch}}$ **do**
  4:        $p(t|X) \leftarrow E_{\text{PS}}(X)$
  5:        **for** $i = 0$ to $k - 1$ **do**
  6:            **if** $i \neq t$ **then**
  7:               $ps_i \leftarrow p(t|X)_i$
  8:               $X_{\text{matched}} \leftarrow$ get closest match to propensity score $ps_i$ with treatment $i$ from $X_{\text{train}}$
  9:               Add sample $X_{\text{matched}}$ to $X_{\text{out}}$
10:      Add $X$ to $X_{\text{out}}$

---

**Model Selection.** Besides accounting for the treatment assignment bias, the other major issue in learning for counterfactual inference from observational data is that, given multiple models, it is not trivial to decide which one to select. The root problem is that we do not have direct access to the true error in estimating counterfactual outcomes, only the error in estimating the observed factual outcomes. This makes it difficult to perform parameter and hyperparameter optimisation, as we do not know which models are better than others for counterfactual inference on a given dataset. To rectify this problem, we use a nearest neighbour approximation $\hat{\epsilon}_{\text{NN-PEHE}}$ of the $\hat{\epsilon}_{\text{PEHE}}$ metric for the binary (Shalit et al. (2017)) and multiple treatment settings for model selection. The $\hat{\epsilon}_{\text{NN-PEHE}}$ estimates the treatment effect of a given sample by substituting the true counterfactual outcome with the outcome $y_j$ from a respective nearest neighbour NN matched on $X$ using the Euclidean distance.

$$\hat{\epsilon}_{\text{NN-PEHE}} = \frac{1}{N} \sum_{n=0}^{N} \Big( [y_1(\text{NN}(n)) - y_0(\text{NN}(n))] - [\hat{y}_1(n) - \hat{y}_0(n)] \Big)^2 \qquad (4)$$

Analogously to Equations (2) and (3), the $\hat{\epsilon}_{\text{NN-PEHE}}$ metric can be extended to the multiple treatment setting by considering the mean $\hat{\epsilon}_{\text{NN-PEHE}}$ between all $\binom{k}{2}$ possible pairs of treatments (Appendix E).

**Model Architecture.** No different from other applications, the chosen architecture plays a key role in the performance of neural networks when attempting to learn representations for counterfactual inference. Shalit et al. (2017) claimed that the naïve approach of appending the treatment index $t_j$ may perform poorly if $X$ is high-dimensional, because the influence of $t_j$ on the hidden layers may be lost during training. Shalit et al. (2017) subsequently introduced the TARNET architecture to rectify this issue. Since the original TARNET was limited to the binary treatment setting, we extended the TARNET architecture to the multiple treatment setting (Figure 1). We did so by using $k$ head networks, one for each treatment option over a set of shared base layers, each with $L$ layers. In TARNET, the $j$th head network is only trained on samples from treatment $t_j$. The shared layers are trained on all samples. By using a head network for each treatment, we ensure $t_j$ maintains an appropriate degree of influence on the network output at all points during training.

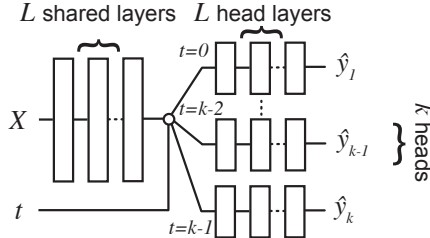

Figure 1: The TARNET architecture with $k$ heads for the multiple treatment setting.

Table 2: Comparisons of the datasets used in our experiments. We evaluate on three semi-synthetic datasets with varying numbers of treatments and samples, and a real-world clinical trial dataset.

| Dataset | Type | # Samples | # Features | # Treatments | Counterfactuals |
|---|---|---|---|---|---|
| IHDP | semi-synthetic | 747 | 25 | 2 | available |
| Jobs | real-world | 3212 | 7 | 2 | not available |
| News | semi-synthetic | 5000 | 2870 | 2/4/8/16 | available |
| TCGA | semi-synthetic | 9659 | 20531 | 4 | available |

## 4 EXPERIMENTS

Our experiments aimed to answer the following questions:

1. What is the comparative performance of PM in inferring counterfactual outcomes in the binary and multiple treatment setting compared to existing state-of-the-art methods?
2. Does model selection by NN-PEHE outperform selection by factual MSE?
3. How does the relative number of matched samples within a minibatch affect performance?
4. How well does PM cope with an increasing treatment assignment bias in the observed data?
5. How does the presence of hidden confounders influence the performance of PM?
6. How do the learning dynamics of minibatch matching compare to dataset-level matching?

### 4.1 DATASETS

We performed experiments on four real-world and semi-synthetic datasets (Table 2) with binary and multiple treatment options in order to gain a better understanding of the empirical properties of PM.

**Infant Health and Development Program (IHDP).** The IHDP dataset (Hill (2011)) contains data from a randomised study on the impact of specialist visits on the cognitive development of children, and consists of 747 children with 25 covariates describing properties of the children and their mothers. Children that did not receive specialist visits were part of a control group. The outcomes were simulated using the NPCI package from Dorie (2016). The IHDP dataset is biased because the treatment groups had a biased subset of the treated population removed (Shalit et al. (2017)). We used the same simulated outcomes[2] as Shalit et al. (2017).

**Jobs.** The Jobs dataset (LaLonde (1986)) is a blend of data from randomised and observational studies on the effect of professional training programs on unemployment, and is a commonly used benchmark dataset for counterfactual inference in the binary setting. Each sample consisted of demographic covariates, such as age, gender and previous income, and the treatment was enrolment in a job training program. All samples that were not enrolled in a training program were considered control cases. The task was to predict the potential unemployment given the covariates and treatment. We used the same feature and sample set as described in Dehejia & Wahba (2002); Smith & Todd (2005); Shalit et al. (2017). A subgroup of the data was randomised and we therefore were able to estimate the ground truth average treatment effect on the treated (ATT) (Shalit et al. (2017)). Treatment assignment in Jobs is biased because parts of the data come from an observational study.

---

[2]Available at: `http://www.mit.edu/~fredrikj/files/IHDP-1000.tar.gz`

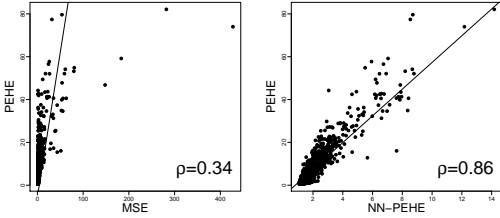
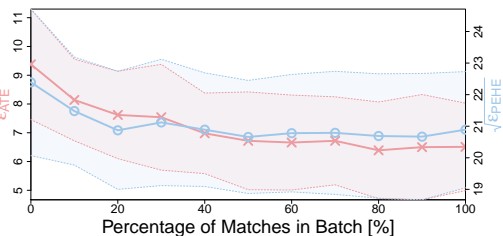

Figure 2: Correlation analysis of the real PEHE (y-axis) with the mean squared error (MSE; left) and the nearest neighbour approximation of the precision in estimation of heterogenous effect (NN-PEHE; right) across over 20'000 model evaluations on the validation set of IHDP. Scatterplots show a subsample of 1'400 data points. $\rho$ indicates the Pearson correlation. NN-PEHE correlates significantly better with PEHE than MSE.

Figure 3: Change in error (y-axes) in terms of precision in estimation of heterogenous effect (PEHE) and average treatment effect (ATE) when increasing the percentage of matches in each minibatch (x-axis). Symbols correspond to the mean value of $\hat{\epsilon}_{ATE}$ (red) and $\sqrt{\hat{\epsilon}_{PEHE}}$ (blue) on the test set of News-8 across 50 repeated runs with new outcomes (lower is better). Performance improves with more matches added.

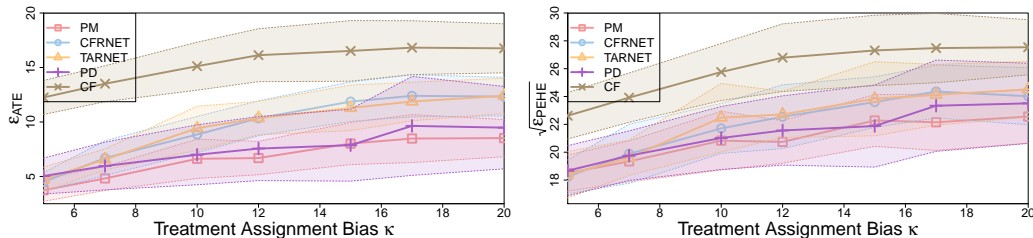

Figure 4: Comparison of several state-of-the-art methods for counterfactual inference on the test set of the News-8 dataset when varying the treatment assignment imbalance $\kappa$ (x-axis), i.e. how much the treatment assignment is biased towards more effective treatments. Symbols correspond to the mean value of $\hat{\epsilon}_{\text{ATE}}$ (left) and $\sqrt{\hat{\epsilon}_{\text{PEHE}}}$ (right) across 50 repeated runs with new outcomes (lower is better). The shaded area indicates the standard deviation. PM handles an increasing treatment assignment bias $\kappa$ better than existing state-of-the-art methods in terms of both $\hat{\epsilon}_{\text{ATE}}$ and $\sqrt{\hat{\epsilon}_{\text{PEHE}}}$.

**News.** The News dataset was first proposed as a benchmark for counterfactual inference by Johansson et al. (2016) and consists of 5000 randomly sampled news articles from the NY Times corpus[3]. The News dataset contains data on the opinion of media consumers on news items. The samples $X$ represent news items consisting of word counts $x_i \in \mathbb{N}$, the outcome $y_j \in \mathbb{R}$ is the reader's opinion of the news item, and the $k$ available treatment options represent various devices that could be used for viewing, e.g. smartphone, tablet, desktop, television or others (Johansson et al. (2016)). We extended the original dataset specification in (Johansson et al. (2016)) to enable the simulation of arbitrary numbers of viewing devices. To model that consumers prefer to read certain media items on specific viewing devices, we train a topic model on the whole NY Times corpus and define $z(X)$ as the topic distribution of news item $X$. We then randomly pick $k + 1$ centroids in topic space, with $k$ centroids $z_j$ per viewing device and one control centroid $z_c$. We assigned a random Gaussian outcome distribution with mean $\mu_j \sim \mathcal{N}(0.45, 0.15)$ and standard deviation $\sigma_j \sim \mathcal{N}(0.1, 0.05)$ to each centroid. For each sample, we drew ideal potential outcomes from that Gaussian outcome distribution $\tilde{y}_j \sim \mathcal{N}(\mu_j, \sigma_j) + \epsilon$ with $\epsilon \sim \mathcal{N}(0, 0.15)$. We then defined the unscaled potential outcomes $\bar{y}_j = \tilde{y}_j * [\text{D}(z(X), z_j) + \text{D}(z(X), z_c)]$ as the ideal potential outcomes $\tilde{y}_j$ weighted by the sum of distances to centroids $z_j$ and the control centroid $z_c$ using the Euclidean distance as distance D. We assigned the observed treatment $t$ using $t|x \sim \text{Bern}(\text{softmax}(\kappa \bar{y}_j))$ with a treatment assignment bias coefficient $\kappa$, and the true potential outcome $y_j = C\bar{y}_j$ as the unscaled potential outcomes $\bar{y}_j$ scaled by a coefficient $C = 50$. We used four different variants of this dataset with $k = 2, 4, 8$, and 16 viewing devices, and $\kappa = 10, 10, 10$, and 7, respectively. Higher values of $\kappa$ indicate a higher expected treatment assignment bias depending on $\bar{y}_j$. $\kappa = 0$ indicates no assignment bias.

**The Cancer Genomic Atlas (TCGA).** The TCGA project collected gene expression data from various types of cancers in 9659 individuals (Weinstein et al. (2013)). There were four available clinical treatment options: (1) medication, (2) chemotherapy, (3) surgery, or (4) both surgery and chemotherapy. We used a synthetic outcome function that simulated the risk of cancer recurrence after receiving either of the treatment options based on the real-world gene expression data. Before further processing, we standardised the gene expression data using the mean and standard deviations of gene expression at each gene locus for normal tissue in the training set. To produce the potential outcomes $Y$, we first selected $k + 1$ gene expression phenotypes as representative phenotypes: One phenotype $X_j$ for each treatment option and one as a control phenotype $X_c$. Analogously to the News dataset, we assigned a random Gaussian outcome distribution with mean $\mu_j \sim \mathcal{N}(0.45, 0.15)$ and standard deviation $\sigma_t \sim \mathcal{N}(0.1, 0.05)$ to each phenotype. For each sample, we drew ideal potential outcomes from that Gaussian outcome distribution $\tilde{y}_j \sim \mathcal{N}(\mu_j, \sigma_j) + \epsilon$ with $\epsilon \sim \mathcal{N}(0, 0.15)$. We then defined the unscaled potential outcomes $\bar{y}_j = \tilde{y}_j * [\text{D}(X, X_{t_j}) + \text{D}(X, X_c)]$ as the ideal potential outcomes $\tilde{y}_j$ weighted by the sum of distances to phenotype $X_{t_j}$ and the control phenotype $X_c$ using the cosine similarity as distance metric D. We assigned the observed treatment $t$ using $t|x \sim \text{Bern}(\text{softmax}(\kappa \bar{y}_j))$ with a treatment assignment bias coefficient $\kappa = 10$, and the true potential outcome $y_j = C\bar{y}_j$ as the unscaled potential outcomes $\bar{y}_j$ scaled by a coefficient $C = 50$.

All datasets with the exception of IHDP were split into a training (63%), validation (27%) and test set (10% of samples). For IHDP we used exactly the same splits as previously used by Shalit et al. (2017). We repeated experiments on IHDP, Jobs, News and TCGA 1000, 10, 50, and 5 times, respectively. We reassigned outcomes and treatments with a new random seed for each repetition.

---

[3]https://archive.ics.uci.edu/ml/datasets/bag+of+words

Table 3: Comparison of methods for counterfactual inference with two available treatment options on IHDP, Jobs and News-2. We report the mean value $\pm$ the standard deviation of $\sqrt{\epsilon_{\text{PEHE}}}$, $\epsilon_{\text{ATE}}$, $R_{\text{Pol}}$ and $\epsilon_{\text{ATT}}$ on the test sets over a number of repeated runs. Where available we list the numbers reported by the original authors. n.r. = not reported. $\dagger$ = significantly different from PM ($\alpha < 0.05$)

| Method | IHDP (1000 repeats) | | Jobs (10 repeats) | | News-2 (50 repeats) | |
|---|---|---|---|---|---|---|
| | $\sqrt{\epsilon_{\text{PEHE}}}$ | $\epsilon_{\text{ATE}}$ | $R_{\text{Pol}}(\pi)$ | $\epsilon_{\text{ATT}}$ | $\sqrt{\hat{\epsilon}_{\text{PEHE}}}$ | $\hat{\epsilon}_{\text{ATE}}$ |
| PM | $0.84 \pm 0.61$ | $0.24 \pm 0.20$ | $0.18 \pm 0.10$ | $0.16 \pm 0.07$ | $\mathbf{16.76} \pm 1.26$ | $\mathbf{3.99} \pm 1.01$ |
| + on $X$ | $0.81 \pm 0.57$ | $0.24 \pm 0.20$ | $0.23 \pm 0.11$ | $0.19 \pm 0.09$ | $17.06 \pm 1.22$ | $4.14 \pm 1.27$ |
| + MLP | $0.83 \pm 0.57$ | $0.23 \pm 0.20$ | $0.19 \pm 0.12$ | $0.16 \pm 0.07$ | $\dagger\,18.38 \pm 1.46$ | $\dagger\,5.90 \pm 2.07$ |
| kNN | $\dagger\,4.1 \pm 0.2$ | $\dagger\,0.79 \pm 0.05$ | $\dagger\,0.26 \pm 0.0$ | $0.13 \pm 0.05$ | $\dagger\,18.14 \pm 1.64$ | $\dagger\,7.83 \pm 2.55$ |
| PSM$_{\text{PM}}$ | n.r. | n.r. | $0.18 \pm 0.12$ | $0.18 \pm 0.09$ | $\dagger\,17.49 \pm 1.49$ | $\dagger\,5.02 \pm 2.34$ |
| PSM$_{\text{MI}}$ | $\dagger\,2.70 \pm 3.85$ | $\dagger\,0.49 \pm 0.81$ | $0.20 \pm 0.09$ | $\dagger\,0.42 \pm 0.22$ | $\dagger\,17.40 \pm 1.30$ | $\dagger\,4.89 \pm 2.39$ |
| RF | $\dagger\,6.6 \pm 0.3$ | $\dagger\,0.96 \pm 0.06$ | $\dagger\,0.28 \pm 0.0$ | $\dagger\,0.09 \pm 0.04$ | $\dagger\,17.39 \pm 1.24$ | $\dagger\,5.50 \pm 1.20$ |
| CF | $\dagger\,3.8 \pm 0.2$ | $\dagger\,0.40 \pm 0.03$ | $0.20 \pm 0.0$ | $\dagger\,0.07 \pm 0.03$ | $\dagger\,18.36 \pm 1.73$ | $\dagger\,8.48 \pm 2.46$ |
| BART | $\dagger\,2.3 \pm 0.1$ | $\dagger\,0.34 \pm 0.02$ | $\dagger\,0.25 \pm 0.0$ | $\dagger\,0.08 \pm 0.03$ | $\dagger\,18.53 \pm 2.02$ | $\dagger\,5.40 \pm 1.53$ |
| GANITE | $\dagger\,2.4 \pm 0.4$ | $\dagger\,0.49 \pm 0.05$ | $\mathbf{0.14} \pm 0.01$ | $\dagger\,0.06 \pm 0.03$ | $\dagger\,18.28 \pm 1.66$ | $\dagger\,4.65 \pm 2.12$ |
| BNN | $\dagger\,2.1 \pm 0.1$ | $\dagger\,0.42 \pm 0.03$ | $0.24 \pm 0.0$ | $\dagger\,0.09 \pm 0.04$ | n.r. | n.r. |
| PD | n.r. | n.r. | n.r. | n.r. | $\dagger\,17.52 \pm 1.62$ | $4.69 \pm 3.17$ |
| TARNET | $\dagger\,0.95 \pm 0.02$ | $\dagger\,0.28 \pm 0.01$ | $0.21 \pm 0.0$ | $0.11 \pm 0.04$ | $17.17 \pm 1.25$ | $\dagger\,4.58 \pm 1.29$ |
| CFRNET$_{\text{Wass}}$ | $\dagger\,\mathbf{0.76} \pm 0.02$ | $\dagger\,0.27 \pm 0.01$ | $0.21 \pm 0.0$ | $\dagger\,0.09 \pm 0.03$ | $16.93 \pm 1.12$ | $\dagger\,4.54 \pm 1.48$ |
| CEVAE | $\dagger\,2.7 \pm 0.1$ | $\dagger\,0.46 \pm 0.02$ | $\dagger\,0.26 \pm 0.0$ | $\dagger\,\mathbf{0.03} \pm 0.01$ | n.r. | n.r. |
| CMGP | $\dagger\,0.77 \pm 0.11$ | $\dagger\,\mathbf{0.13} \pm 0.12$ | $0.24 \pm 0.05$ | $\dagger\,0.09 \pm 0.07$ | n.r. | n.r. |

## 4.2 EXPERIMENTAL SETUP

**Models.** We evaluated PM, ablations, baselines, and all relevant state-of-the-art methods: kNN (Ho et al. (2007)), BART (Chipman et al. (2010); Chipman & McCulloch (2016)), Random Forests (RF) (Breiman (2001)), CF (Wager & Athey (2017)), GANITE (Yoon et al. (2018)), Balancing Neural Network (BNN) (Johansson et al. (2016)), TARNET (Shalit et al. (2017)), Counterfactual Regression Network using the Wasserstein regulariser (CFRNET$_{\text{Wass}}$) (Shalit et al. (2017)), CEVAE (Louizos et al. (2017)), CMGP (Alaa & van der Schaar (2017)), and PD (Alaa et al. (2017)). We trained a Support Vector Machine (SVM) with probability estimation (Pedregosa et al. (2011)) to estimate $p(t|X)$ for PM on the training set. We also evaluated preprocessing the entire training set with PSM using the same matching routine as PM (PSM$_{\text{PM}}$) and the "MatchIt" package (PSM$_{\text{MI}}$, Ho et al. (2011)) before training a TARNET (Appendix F). In addition, we trained an ablation of PM where we matched on the covariates $X$ (+ on $X$) directly, if $X$ was low-dimensional ($p < 200$), and on a 50-dimensional representation of $X$ obtained via principal components analysis (PCA), if $X$ was high-dimensional, instead of on the propensity score. We also evaluated PM with a multi-layer perceptron (+ MLP) that received the treatment index $t_j$ as an input instead of using a TARNET.

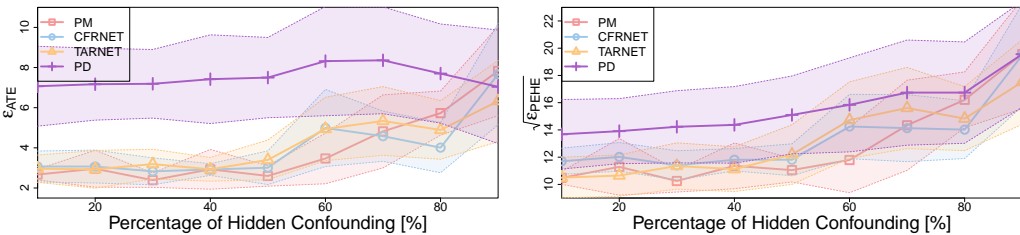

Figure 5: Comparison of several state-of-the-art methods for counterfactual inference on the test set of the TCGA dataset when varying the percentage of hidden confounding (x-axis), i.e. what percentage of the features that were relevant for treatment assignment are visible to the model. Symbols correspond to the mean value of $\hat{\epsilon}_{\text{ATE}}$ (left) and $\sqrt{\hat{\epsilon}_{\text{PEHE}}}$ (right) across 5 repeated runs with new outcomes (lower is better). The shaded area indicates the standard deviation. PM, TARNET and CFRNET are relatively stable up to 60% of hidden confounding. The methods degrade strongly in counterfactual prediction performance at levels of hidden confounding higher than 60%.

Table 4: Comparison of methods for counterfactual inference with more than two available treatment options on the News-4, News-8 and News-16. Numbers are mean value $\pm$ the standard deviation of $\sqrt{\hat{\epsilon}_{\text{mPEHE}}}$ and $\hat{\epsilon}_{\text{mATE}}$ over 50 repeated runs. $\dagger$ = significantly different from PM ($\alpha < 0.05$)

| Method | News-4 ($\kappa = 10$) | | News-8 ($\kappa = 10$) | | News-16 ($\kappa = 7$) | |
| --- | --- | --- | --- | --- | --- | --- |
| | $\sqrt{\hat{\epsilon}_{\text{mPEHE}}}$ | $\hat{\epsilon}_{\text{mATE}}$ | $\sqrt{\hat{\epsilon}_{\text{mPEHE}}}$ | $\hat{\epsilon}_{\text{mATE}}$ | $\sqrt{\hat{\epsilon}_{\text{mPEHE}}}$ | $\hat{\epsilon}_{\text{mATE}}$ |
| PM | $21.58 \pm 2.58$ | $10.04 \pm 2.71$ | $\mathbf{20.76 \pm 1.86}$ | $6.51 \pm 1.66$ | $\mathbf{20.24 \pm 1.46}$ | $5.76 \pm 1.33$ |
| + on $X$ | $21.41 \pm 1.75$ | $\dagger\ 8.91 \pm 2.00$ | $20.90 \pm 2.07$ | $\mathbf{6.36 \pm 1.65}$ | $20.67 \pm 1.42$ | $\mathbf{5.73 \pm 1.16}$ |
| + MLP | $\dagger\ 25.05 \pm 2.80$ | $\dagger\ 14.92 \pm 3.14$ | $\dagger\ 24.88 \pm 1.98$ | $\dagger\ 14.00 \pm 2.45$ | $\dagger\ 27.05 \pm 2.47$ | $\dagger\ 16.90 \pm 2.01$ |
| kNN | $\dagger\ 27.92 \pm 2.44$ | $\dagger\ 19.40 \pm 3.12$ | $\dagger\ 26.20 \pm 2.18$ | $\dagger\ 15.11 \pm 2.34$ | $\dagger\ 27.64 \pm 2.40$ | $\dagger\ 17.27 \pm 2.10$ |
| PSM$_{\text{PM}}$ | $\dagger\ 22.74 \pm 2.58$ | $\dagger\ 11.62 \pm 2.69$ | $\dagger\ 22.16 \pm 1.79$ | $\dagger\ 8.81 \pm 1.96$ | $\dagger\ 23.57 \pm 2.48$ | $\dagger\ 11.04 \pm 2.50$ |
| PSM$_{\text{MI}}$ | $\dagger\ 37.26 \pm 2.28$ | $\dagger\ 30.19 \pm 2.47$ | $\dagger\ 30.50 \pm 1.70$ | $\dagger\ 22.09 \pm 1.98$ | $\dagger\ 28.17 \pm 2.02$ | $\dagger\ 18.81 \pm 1.74$ |
| RF | $\dagger\ 26.59 \pm 3.02$ | $\dagger\ 18.03 \pm 3.18$ | $\dagger\ 23.77 \pm 2.14$ | $\dagger\ 12.40 \pm 2.29$ | $\dagger\ 26.13 \pm 2.48$ | $\dagger\ 15.91 \pm 2.00$ |
| CF | $\dagger\ 27.28 \pm 2.59$ | $\dagger\ 19.04 \pm 3.15$ | $\dagger\ 25.76 \pm 2.05$ | $\dagger\ 15.12 \pm 2.20$ | $\dagger\ 26.85 \pm 2.29$ | $\dagger\ 17.03 \pm 1.99$ |
| BART | $\dagger\ 26.41 \pm 3.10$ | $\dagger\ 17.14 \pm 3.51$ | $\dagger\ 25.78 \pm 2.66$ | $\dagger\ 14.80 \pm 2.56$ | $\dagger\ 27.45 \pm 2.84$ | $\dagger\ 17.50 \pm 2.49$ |
| GANITE | $\dagger\ 24.50 \pm 2.27$ | $\dagger\ 13.84 \pm 2.69$ | $\dagger\ 23.58 \pm 2.48$ | $\dagger\ 11.20 \pm 2.84$ | $\dagger\ 25.12 \pm 3.53$ | $\dagger\ 13.20 \pm 3.28$ |
| PD | $\mathbf{20.88 \pm 3.24}$ | $\dagger\ \mathbf{8.47 \pm 4.51}$ | $21.19 \pm 2.29$ | $7.29 \pm 2.97$ | $\dagger\ 22.28 \pm 2.25$ | $\dagger\ 10.65 \pm 2.22$ |
| TARNET | $\dagger\ 23.40 \pm 2.20$ | $\dagger\ 13.63 \pm 2.18$ | $\dagger\ 22.39 \pm 2.32$ | $\dagger\ 9.38 \pm 1.92$ | $\dagger\ 21.19 \pm 2.01$ | $\dagger\ 8.30 \pm 1.66$ |
| CFRNET$_{\text{Wass}}$ | $\dagger\ 22.65 \pm 1.97$ | $\dagger\ 12.96 \pm 1.69$ | $\dagger\ 21.64 \pm 1.82$ | $\dagger\ 8.79 \pm 1.68$ | $\dagger\ 20.87 \pm 1.46$ | $\dagger\ 8.05 \pm 1.40$ |

**Architectures.** To ensure that differences between methods of learning counterfactual representations for neural networks are not due to differences in architecture, we based the neural architectures for TARNET, CFRNET$_{\text{Wass}}$, PD and PM on the same, previously described extension of the TARNET architecture (Shalit et al. (2017), Details in Appendix G) to the multiple treatment setting.

**Hyperparameters.** For the IHDP, Jobs, News and TCGA dataset we respectively used 30, 30, 10, and 5 optimisation runs for each method using randomly selected hyperparameters from predefined ranges (Appendix H). We selected the best model across the runs based on validation set $\hat{\epsilon}_{\text{NN-PEHE}}$.

**Metrics.** We calculated the $\epsilon_{\text{PEHE}}$ and $\epsilon_{\text{ATE}}$ for datasets for which we know the outcome generating process. For the Jobs dataset, we did not have access to counterfactual outcomes. We therefore calculated the ATT $= \frac{1}{|T_1 \cap E|} \sum_{n=0}^{N} y_1(n) - \frac{1}{|T_0 \cap E|} \sum_{n=0}^{N} y_0(n)$ where $E$ was the randomised subset of the dataset, and reported the error in estimating ATT $\epsilon_{\text{ATT}} = |\text{ATT} - \frac{1}{|T_1 \cap E|} \sum_{n=0}^{N} \hat{y}_1(n)|$, and the policy risk $R_{\text{Pol}}(\pi)$ (Shalit et al. (2017); Yoon et al. (2018)). The policy risk is the average loss in value when treating according to the policy $\pi$ implied by an ITE estimator (Appendix I).

## 5 RESULTS AND DISCUSSION

**Counterfactual Inference.** We evaluated the counterfactual inference performance of the listed models in the binary setting (Table 3) and multiple treatment setting (Table 4). On IHDP, PM reached the third best performance in terms of $\sqrt{\epsilon_{\text{PEHE}}}$ after CFRNET and CMGP, and the second best $\epsilon_{\text{ATE}}$ after CMGP. On Jobs, PM reached the second best $R_{\text{Pol}}(\pi)$ after GANITE. On the binary News-2, PM outperformed all other methods in terms of $\sqrt{\epsilon_{\text{PEHE}}}$ and $\epsilon_{\text{ATE}}$. On the News-4/8/16 datasets with more than two treatments, PM consistently outperformed all other methods - in some cases by a large margin - on both metrics with the exception of the News-4 dataset, where PM came second to PD. The strong performance of PM across a wide range of datasets with varying amounts of treatment options is remarkable considering how simple it is compared to other, highly specialised methods. Notably, PM consistently outperformed both CFRNET, with the exception of the IHDP dataset, which accounted for covariate imbalances between treatments via regularisation rather than matching, and PSM$_{\text{MI}}$, which accounted for covariate imbalances by preprocessing the entire training set with a matching algorithm (Ho et al. (2011)). We also found that matching on the propensity score was, in almost all cases, not significantly different from matching on $X$ directly when $X$ was low-dimensional, or a low-dimensional representation of $X$ when $X$ was high-dimensional (+ on $X$). This indicates that PM is effective with any low-dimensional balancing score. In addition, using PM with the TARNET architecture outperformed the MLP (+ MLP) in almost all cases, with the exception of the low-dimensional IHDP and Jobs. We conclude that matching on the propensity score or a low-dimensional representation of $X$ and using the TARNET architecture are sensible default configurations, particularly when $X$ is high-dimensional.

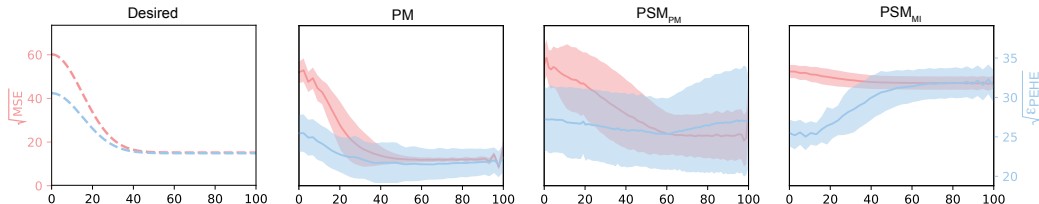

Figure 6: Comparison of the learning dynamics during training (normalised training epochs; from start = 0 to end = 100 of training, x-axis) of several matching-based methods on the validation set of News-8. The coloured lines correspond to the mean value of the factual error ($\sqrt{\text{MSE}}$; red) and the counterfactual error ($\sqrt{\hat{\epsilon}_{\text{PEHE}}}$; blue) across 10 randomised hyperparameter configurations (lower is better). The shaded area indicates the standard deviation. All methods used exactly the same model and hyperparameters and only differed in how they addressed treatment assignment imbalance. The leftmost figure shows the desired behavior of the counterfactual and factual error jointly decreasing until convergence. PM exhibits the desired behavior of aligning the optimisation of the factual and counterfactual error. $\text{PSM}_{\text{PM}}$ shows a weaker degree of alignment and much higher variance. In contrast, $\text{PSM}_{\text{MI}}$ shows the undesired behavior of the counterfactual error increasing as the factual error decreases, indicating that the models were overfitting to the properties of the treated group.

**Model Selection.** To judge whether NN-PEHE is more suitable for model selection for counterfactual inference than MSE, we compared their respective correlations with the PEHE on IHDP. We found that NN-PEHE correlates significantly better with the PEHE than MSE (Figure 2).

**Number of Matches per Minibatch.** To determine the impact of matching fewer than 100% of all samples in a batch, we evaluated PM on News-8 trained with varying percentages of matched samples on the range 0 to 100% in steps of 10% (Figure 3). We found that including more matches indeed consistently reduces the counterfactual error up to 100% of samples matched. Interestingly, we found a large improvement over using no matched samples even for relatively small percentages (<40%) of matched samples per batch. This shows that propensity score matching within a batch is indeed highly effective at improving the training of neural networks for counterfactual inference.

**Treatment Assignment Bias.** To assess how the predictive performance of the different methods is influenced by increasing amounts of treatment assignment bias, we evaluated their performances on News-8 while varying the assignment bias coefficient $\kappa$ on the range of 5 to 20 (Figure 4). We found that PM handles high amounts of assignment bias better than existing state-of-the-art methods.

**Hidden Confounding.** To analyse the sensitivity of the different methods to hidden confounding, we successively retrained the models on a reduced set of covariates $x_i$ in increasing steps of 10% (= 2053 gene loci) in the range of 10% to 90% (Figure 5). We removed the covariates from the model but kept them for computing assignments and outcomes, i.e. they were hidden confounders. The results showed that PM was competitive with TARNET and CFRNET in terms of robustness to hidden confounding. Notably, PM, TARNET and CFRNET were relatively stable up to levels of 60% of hidden confounding, likely because some gene loci are highly correlated.

**Comparing Minibatch and Dataset Matching.** As outlined previously, it is possible to optimise the counterfactual error using the observed factual samples if (1) we manage to break the dependence of treatment assignment on $X$, and (2) we observe all relevant variables (Ho et al. (2007)). If we were successful in balancing the covariates, we would expect that the counterfactual error is implicitly and consistently improved alongside the factual error, despite optimising the neural network for the observed factual loss only. To elucidate to what degree this is the case when using the various evaluated matching-based methods, we evaluated the respective training dynamics of PM, $\text{PSM}_{\text{PM}}$ and $\text{PSM}_{\text{MI}}$ (Figure 6). We found that PM better conforms to the desired behavior than $\text{PSM}_{\text{PM}}$ and $\text{PSM}_{\text{MI}}$. $\text{PSM}_{\text{PM}}$, which used the same matching strategy as PM but on the dataset level, showed a much higher variance than PM. $\text{PSM}_{\text{MI}}$ was overfitting to the treated group.

**Limitations.** A limitation of this work is that the theory of matching to balance the covariates only holds in the strongly ignorable setting, i.e. under the assumption that there are no unobserved confounders. However, our experiments (Figure 5) and related works showed that even the presence of large amounts of hidden confounders may not necessarily decrease the performance of ITE estimators in practice if we observe proxy variables (Montgomery et al. (2000); Louizos et al. (2017)). In future work, we would like to integrate PM with diagnostic tools that address hidden confounding (Kallus & Zhou (2018)), and the ability to learn from multiple related tasks (Schwab et al. (2018)).

# 6    CONCLUSION

We presented PM, a new and simple method for training neural networks for estimating ITEs from observational data that extends to any number of available treatment options. In addition, we extended the PEHE metric to settings with more than two treatments, and introduced a nearest neighbour approximation of PEHE and mPEHE that can be used for model selection without having access to counterfactual outcomes. We performed experiments on several real-world and semi-synthetic datasets that showed that PM outperforms a number of more complex state-of-the-art methods in inferring counterfactual outcomes. We also found that the NN-PEHE correlates significantly better with real PEHE than MSE, that including more matched samples in each minibatch improves the learning of counterfactual representations, and that PM handles an increasing treatment assignment bias better than existing state-of-the-art methods. PM may be used for settings with any amount of treatment options, is compatible with any existing neural network architecture, simple to implement, and does not introduce any additional hyperparameters or computational complexity.

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

# SUPPLEMENTARY MATERIAL FOR:
# "PERFECT MATCH: A SIMPLE METHOD FOR LEARNING REPRESENTATIONS FOR COUNTERFACTUAL INFERENCE WITH NEURAL NETWORKS"

**Anonymous authors**

## A   PRECISION IN ESTIMATION OF HETEROGENOUS EFFECT (PEHE)

When the underlying noiseless distributions $\mu_j$ are not given, PEHE is calculated using:

$$\hat{\epsilon}_{\text{PEHE}} = \frac{1}{N} \sum_{n=0}^{N} \Big( [y_1(n) - y_0(n)] - [\hat{y}_1(n) - \hat{y}_0(n)] \Big)^2 \tag{A.1}$$

## B   AVERAGE TREATMENT EFFECT (ATE)

The ATE measures the average difference in effect across the whole population (Hill (2011)). It can be useful indicator of how well an ITE estimator performs at comparing two treatments across the entire population.

$$\epsilon_{\text{ATE}} = || \frac{1}{N} \sum_{n=0}^{N} \mathbb{E}_{y(n) \sim \mu(n)}[y(n)] - \frac{1}{N} \sum_{n=0}^{N} \hat{y}(n) ||_2^2 \tag{B.1}$$

Similar to Equation (A.1), the counterfactual treatment effect can be estimated using the noisy ground truth outcomes if the underlying noiseless distributions $\mu_j$ are not given:

$$\hat{\epsilon}_{\text{ATE}} = || \frac{1}{N} \sum_{n=0}^{N} y(n) - \frac{1}{N} \sum_{n=0}^{N} \hat{y}(n) ||_2^2 \tag{B.2}$$

## C   DETAILS PERFECT MATCH (PM)

Algorithm 1 outlines the procedure of using PM to augment a batch with propensity score matches for training a neural network with stochastic gradient descent. We trained a Support Vector Machine (SVM) with probability estimation as propensity score estimator $E_{\text{PS}}$. To speed up recalling nearest neighbours, we additionally prepared an index per treatment into $X_{\text{train}}$ sorted by propensity score. At augmentation time, we used binary search on this index to find the nearest neighbours by propensity score in O(log $N_t$) where $N_t$ is the number of samples $X$ in $X_{\text{train}}$ assigned to the treatment group $t$. To avoid overfitting to specific edge samples when propensity scores are not distributed evenly in the training set, we chose at random from one of the $k=6$ closest samples by propensity score. See Table S1 for a comparison of the effects on predictive performance of choosing varying values of $k$. For the "+ on $X$" model we matched, using the Euclidean distance, directly on the covariates $X$ on IHDP and Jobs, and using a low dimensional representation of $X$, obtained using principal component analysis (PCA) with 50 principal components, on News.

Table S1: Comparison of PM with varying numbers of nearest neighbours $k$ considered for randomised matching on the News-2/4/8/16 datasets ($k$=1 corresponds to no randomisation; $k$=6, in cursive, are the results reported in the main body). We report the mean value $\pm$ the standard deviation of $\sqrt{\hat{\epsilon}_{\text{PEHE}}}$ on the test sets over 50 runs. There was no significant differences between different levels of $k$ and $k$=6 (significance level $\alpha < 0.05$). In particular, there was no significant difference between using randomisation and not using randomisation.

| Method | News-2 $\sqrt{\hat{\epsilon}_{\text{PEHE}}}$ | News-4 $\sqrt{\hat{\epsilon}_{\text{PEHE}}}$ | News-8 $\sqrt{\hat{\epsilon}_{\text{PEHE}}}$ | News-16 $\sqrt{\hat{\epsilon}_{\text{PEHE}}}$ |
|---|---|---|---|---|
| PM ($k$=1) | $16.79 \pm 1.16$ | $21.86 \pm 2.24$ | $20.70 \pm 1.87$ | $\mathbf{20.06} \pm 1.43$ |
| PM ($k$=3) | $16.82 \pm 1.17$ | $21.49 \pm 2.43$ | $\mathbf{20.50} \pm 1.48$ | $20.28 \pm 1.52$ |
| *PM ($k$=6)* | $\mathbf{16.76} \pm 1.26$ | $21.58 \pm 2.58$ | $20.76 \pm 1.86$ | $20.24 \pm 1.46$ |
| PM ($k$=9) | $16.91 \pm 1.18$ | $21.56 \pm 2.58$ | $20.75 \pm 1.53$ | $20.47 \pm 1.94$ |
| PM ($k$=12) | $17.11 \pm 0.94$ | $21.16 \pm 2.20$ | $21.10 \pm 2.14$ | $20.28 \pm 1.64$ |
| PM ($k$=15) | $16.87 \pm 1.14$ | $\mathbf{20.94} \pm 2.05$ | $20.92 \pm 1.80$ | $20.48 \pm 1.80$ |

## D  REDUCED VARIANCE IS LINKED TO IMPROVED PERFORMANCE IN ESTIMATING TREATMENT EFFECTS

In the standard supervised setting, the variance added by minibatch SGD leads to slower convergence (Csiba & Richtrik (2018)), but will not typically lead to models that perform worse if they are trained until a convergence criterion has been met, e.g. early stopping. It is therefore perhaps surprising that we observe a difference not just in convergence (Figure 6), but also in counterfactual estimation performance (Tables 3 and 4) when comparing models trained with and without batch matching.

A potential cause of the observed difference in treatment effect estimation performance is that we do not have access to the true counterfactual error in observational data to select the best model encountered during training. We therefore have to resort to using the factual error (e.g. factual MSE) or an estimator based on the factual error (e.g. NN-PEHE) to select the best encountered model. However, when the variance during training is high, chances are that we select a model based on the factual error that at this point during the training happens to be suboptimal in terms of counterfactual error. Through this mechanism, added variance during training for counterfactual inference is directly linked to worse expected performance in treatment effect estimation (Figure S1). We do not observe the same behavior in standard supervised learning tasks because we have direct access to the underlying objective, and are therefore able to select the best encountered model regardless of the variance during optimisation, i.e. models that during training do not perform well momentarily will not be selected because their measured error is high. We confirmed experimentally that gradient steps are more likely to be in opposing directions when not using PM (Figure S2).

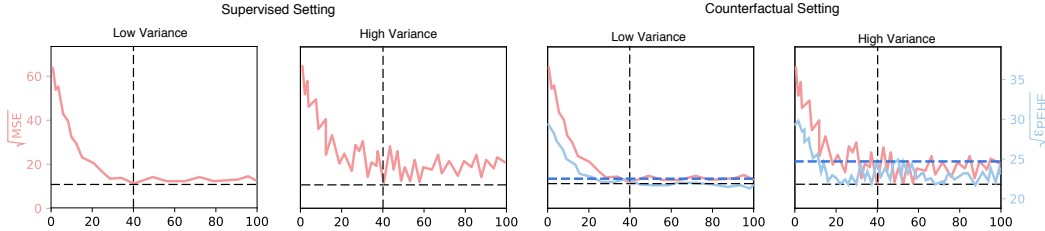

Figure S1: Illustration of the impact of added variance introduced by minibatch SGD during training. In the supervised setting (left), high variance does in general not impact the final performance in terms of expected factual error (red), because we can select (dotted black line) the best encountered model during training using the observed error. In the counterfactual setting (right), we are not able to compute the true counterfactual error (blue) in observational data, and must therefore resort to selecting based on the factual error or a measure derived from the factual error. Even though we encounter models that perform similarly well, we are less likely to select these models under high variance, because we do not have direct access to the counterfactual error and the minimum of the factual error in general does not correspond to the minimum of the counterfactual error, particularly when optimisation of counterfactual and factual error are not well aligned. The dashed blue line indicates the counterfactual error of the model selected based on factual error.

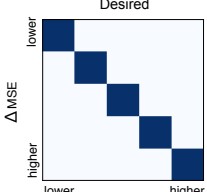 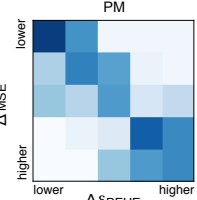 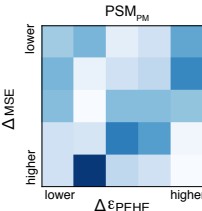 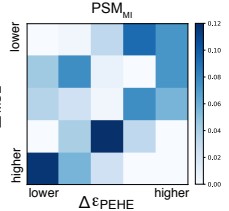

Figure S2: Comparison of the frequencies of individual gradient steps during training of several matching-based methods on the validation set of News-8. The heatmap entries show the frequencies of pairs of differences in factual error ($\Delta$MSE; y-axis) and counterfactual error ($\Delta\epsilon_{\text{PEHE}}$; x-axis) after individual gradient steps. All methods used the same model and only differed in how they addressed treatment assignment bias. The leftmost heatmap shows the desired behavior of gradient steps changing the counterfactual and factual error jointly during training. PM most closely resembles the desired behavior of jointly optimising counterfactual and factual error. In contrast, PSM$_{\text{PM}}$ and PSM$_{\text{MI}}$ have a much higher relative frequency of gradient steps that do not move into the same magnitude or direction (off-diagonal entries) for both factual and counterfactual error.

## E  NEAREST NEIGHBOUR APPROXIMATION OF PEHE FOR MULTIPLE TREATMENTS (NN-MPEHE)

The $\hat{\epsilon}_{\text{NN-PEHE}}$ metric can be extended to the multiple treatment setting by considering the mean $\hat{\epsilon}_{\text{NN-PEHE}}$ between all $\binom{k}{2}$ possible pairs of treatments:

$$\hat{\epsilon}_{\text{NN-mPEHE}} = \frac{1}{\binom{k}{2}} \sum_{i=0}^{k-1} \sum_{j=0}^{i-1} \hat{\epsilon}_{\text{NN-PEHE},i,j} \tag{E.1}$$

## F  PSEUDOCODE PROPENSITY SCORE MATCHING (PSM)

Algorithm S2 outlines the procedure of preprocessing a training set using PSM$_{\text{MI}}$. For PSM$_{\text{PM}}$, we used the PM matching algorithm (see Algorithm 1) to find matches between samples from the treatment groups. For PSM$_{\text{MI}}$, we used the MatchIt package (Ho et al. (2011)) on setting "nearest" to find matches between sets of samples from two treatment groups (function `get_matched_samples`).

---

**Algorithm S2** Preprocessing a training set using Propensity Score Matching (PSM). After preprocessing, the training set contains an equal number of samples from each treatment group and the covariates $x_i$ across all treatment groups are approximately balanced.

---

**Input:** $N$ training samples $X_{\text{train}}$ with assigned treatments $t$, Number of treatment options $k$, function `get_matched_samples` to find nearest matches by propensity score between sets of samples

**Output:** Preprocessed training set $X_{\text{out}}$ consisting of matched samples from each treatment group

1: **procedure** PREPROCESS_TRAINING_SET:
2:     $t_{\text{base}} \leftarrow$ index of treatment with smallest number of samples in $X_{\text{train}}$
3:     $X_{\text{base}} \leftarrow$ all samples in $X_{\text{train}}$ with treatment $t_{\text{base}}$
4:     **for** $i = 0$ to $k - 1$ **do**
5:         **if** $i \neq t_{\text{base}}$ **then**
6:             $X_{\text{current}} \leftarrow$ all samples in $X_{\text{train}}$ with treatment $i$
7:             $X_{\text{matched}} \leftarrow$ `get_matched_samples`$(X_{\text{base}}, X_{\text{current}})$
8:             Add all samples in $X_{\text{matched}}$ to $X_{\text{out}}$
9:     Add all samples in $X_{\text{base}}$ to $X_{\text{out}}$

---

## G TARNET, CFRNET AND PD FOR MULTIPLE TREATMENTS

In TARNET models, we used ELU nonlinearities between the $L$ hidden layers with $M$ hidden units. $L$ and $M$ were hyperparameters that we optimised during hyperparameter optimisation (Section H). We did not use batch normalisation (BN). To extend CFRNET to the multiple treatment setting, we defined the first treatment option as the control treatment and regularised all treatment options to have the same activation distribution in the topmost shared layer (Shalit et al. (2017)). For PD, we only used propensity dropout and the propensity estimation network. We did not make use of the alternating training schedule proposed in (Alaa et al. (2017)). The "+ MLP" model was a simple MLP with $L$ hidden layers of $M$ hidden units that received the treatment option index $t_j$ as an additional input along with the covariates $X$, and output a $k$-dimensional potential outcome vector $\hat{Y}$. The MLP used ELU nonlinearities between the $L$ hidden layers, and also did not use BN.

## H HYPERPARAMETERS

To ensure a fair comparison, we used a standardised approach to hyperparameter optimisation for those methods for which we did not have previously reported performance numbers. In particular, each method we trained was optimised over the same amount of hyperparameter optimisation runs. For the methods that used neural network models (TARNET, CFRNET, PD, $PSM_{PM}$, $PSM_{MI}$, including "+ on $X$" and "+ MLP", and PM), we chose hyperparameters at random from predefined ranges (Table S2). For CFRNET, we additionally varied the weight of the imbalance penalty at random between 0.1, 1.0, and 10.0. All methods that used a neural network model used the TARNET architecture to ensure differences in performance are not due to architectural differences. To train PM for the Jobs dataset, we used fixed hyperparameters: A batch size of 50, 60 hidden units per layer, and 3 hidden layers. For optimisation, we used the Adam optimiser with a learning rate of 0.001 for a maximum of 100 (News, TCGA) or 400 (IHDP, Jobs) with an early stopping patience of 30 on the factual MSE. We used the default hyperparameters for BART and CF from the "bart-Machine" (Kapelner & Bleich (2013)) and "grf" (Athey et al. (2016)) R-packages. For GANITE, we used our own implementation since there was no open source implementation available, and - in addition to the parameters in Table S2 - optimised over the supervised loss weights $\alpha$ and $\beta$ (Yoon et al. (2018)) between 0.1, 1, and 10. For the GANITE generators and discriminators, we used MLP architectures with $L$ hidden layers of $M$ hidden units each.

Table S2: Hyperparameter ranges used in the performed experiments.

| Hyperparameter | IHDP | News/TCGA |
|---|---|---|
| Batch size $B$ | 4, 8, 50, 100 | 50 |
| Number of units per hidden layer $M$ | 50, 100, 200 | 40, 60, 80 |
| Number of hidden layers $L$ | 1, 2, 3 | 2, 3 |

## I POLICY RISK $R_{POL}(\pi)$

We define the policy risk $R_{Pol}(\pi)$ as the expected loss in value when treating according to the policy $\pi$ implied by an ITE estimator (Shalit et al. (2017)):

$$R_{Pol}(\pi) = 1 - (\mathbb{E}[y_1|\pi(X) = 1] \cdot p(\pi = 1) + \mathbb{E}[y_0|\pi(X) = 0] \cdot p(\pi = 0)) \tag{I.1}$$

where the policy $\pi(X)$ for a sample $X$ implied by an ITE estimator that produces potential outcomes $\hat{y}$ is defined as:

$$\pi(X) = \begin{cases} 1 & \text{if } \hat{y}_{1,X} - \hat{y}_{0,X} > \lambda \quad \text{(to treat)} \\ 0 & \text{otherwise} \quad\quad\quad\quad \text{(to not treat)} \end{cases} \tag{I.2}$$

## J  ORIGINS OF PERFORMANCE NUMBERS IN TABLE 3

Where available, performance numbers in Table 3 were taken from the original authors for IHDP and Jobs (Table S3). For News-2/4/8/16 and TCGA, performance numbers were produced by us.

Table S3: Origins of the performance results reported in Table 3. Entries show the reference to the original report of the performance results, or "ours" if we produced the results. n.r. = not reported

| Method | IHDP | Jobs |
|---|---|---|
| PM | ours | ours |
| + on $X$ | ours | ours |
| + MLP | ours | ours |
| kNN | (Shalit et al. 2017) | (Shalit et al. 2017) |
| $PSM_{PM}$ | n.r. | ours |
| $PSM_{MI}$ | ours | ours |
| RF | (Shalit et al. 2017) | (Shalit et al. 2017) |
| CF | (Shalit et al. 2017) | (Shalit et al. 2017) |
| BART | (Shalit et al. 2017) | (Shalit et al. 2017) |
| GANITE | (Yoon et al. 2018) | (Yoon et al. 2018) |
| BNN | (Shalit et al. 2017) | (Shalit et al. 2017) |
| PD | n.r. | n.r. |
| TARNET | (Shalit et al. 2017) | (Shalit et al. 2017) |
| $CFRNET_{Wass}$ | (Shalit et al. 2017) | (Shalit et al. 2017) |
| CEVAE | (Louizos et al. 2017) | (Louizos et al. 2017) |
| CMGP | (Yoon et al. 2018) | (Yoon et al. 2018) |

