# OpenReview forum: "Perfect Match: A Simple Method for Learning Representations For Counterfactual Inference With Neural Networks"
_ICLR.cc/2019/Conference_

### Official Review · AnonReviewer2 · 2018-10-31
**Overall a good paper**

**Rating:** 6
**Confidence:** 4

**Review:**

This paper proposes an augmentation of traditional neural network learning to allow for the inference of causal effects. Specifically, they modify the data sampling procedure of SGD during training to use matched samples that are paired via propensity score matching. Experimental results on a number of dataset show that the proposed methodology is comparable to alternative machine learning based causal inference methods.

Overall, I think this is a nice idea. I have two main concerns:
(1) The use of small batches for matching. Figure 2 does alleviate this concern to an extent, but there is a large literature in statistics and the social sciences on the effect that the quality of matches have on the final causal estimand. It is quite possible that this particular dataset is more amenable to PSM. It is also worth noting that while there is bias reduction shown in figure 2, it is not overwhelming.

(2) The use of propensity scores for matching. One of the insights from the heterogeneous treatment effect literature is that it is not difficult to find cases where the propensity of treatment is identical for two sets of covariates that otherwise do not obey any real balance. This can lead to large biases in the final estimate. Given that PSM is still a relatively widely used practice, I don’t think that its use is a ground for rejection in itself, but given that neural networks are often used to estimate complex causal relations when they are used and this paper is interested in individual treatment effects it is worth noting.

I found the experimental setup to do a very good job in covering large portions of the behavior of the algorithm. The final results are a little underwhelming–the proposed method does not appear to clearly define a new state of the art for the tasks it is applied to–but it is often competitive and the paper presents an interesting idea.

---

> ### Author Response · Authors · 2018-11-15
> **Rebuttal Reviewer 2**
>
> We thank Reviewer 2 for the helpful and constructive feedback. We address the feedback below in order.
>
> R2: "I have two main concerns: (1) The use of small batches for matching. (...)"
>
> We believe R2 may have missed this detail in the manuscript: PM does not use small batches for matching. All samples from the whole training set are used for matching (see p. 3, bottom, and Algorithm 1).
>
> R2: "It is quite possible that this particular dataset is more amenable to PSM."
>
> We evaluated PM on four different datasets (IHDP, Jobs, News-2/4/8/16 and TCGA) that cover a wide range of characteristics (Table 2), and the performance of PM was consistently either state-of-the-art or highly competitive with the respective state-of-the-art (Tables 3 and 4). The consistently strong performance of PM in our experiments shows that PM performs well across multiple datasets with varying characteristics.
>
> R2: "(2) The use of propensity scores for matching. (...)"
>
> We thank R2 for this comment. In the most recent revision of the manuscript, we have updated the "+ on X" ablation to use a low-dimensional representation of X for matching (obtained via PCA, 50 principal components) on the News dataset. The results show that matching on a low-dimensional representation is as effective as matching on the propensity score (Tables 3 and 4; p. 8 last paragraph). This indicates that PM is effective with any low-dimensional balancing score, not just the propensity score.
>
> R2: "The final results are a little underwhelming–the proposed method does not appear to clearly define a new state of the art (...)"
>
> Across the four evaluated datasets, PM consistently achieved either the best or a highly competitive performance in the relevant metrics for ITE estimation (PEHE, R_{Pol}). It is important that a method achieves good results consistently across many different datasets with varying characteristics, because we, in general, do not have access to the counterfactual error in practice, i.e. for a given real-world observational dataset we would not know whether a non-consistent method is operating at its best or its worst.
>
> Finally, independently of performance metrics, we would additionally like to stress that PM is easy to implement, compatible with any architecture, does not add computational complexity or hyperparameters, and extends to any number of treatments. This is in stark contrast to the limitations and complexities added by existing state-of-the-art methods (Table 1).

---

### Official Review · AnonReviewer3 · 2018-11-02
**Simple idea, the presentation of the method and experiment results can be improved**

**Rating:** 5
**Confidence:** 4

**Review:**

Summary:
This paper proposed to extend TARNET (Shalit et al. 2017), a representation learning approach for counterfactual inference, in the following ways.

First, to extend TARNET to multiple treatment setting, k head networks (instead of 2) were constructed following the shared MLP layers, where each head network modeled the outcome of one treatment. This extension seemed quite straightforward.

Second, during training, for every sample in a minibatch, find its nearest neighbors from all other treatments and add them to the minibatch. The distance was measured by the propensity score, which was defined the probability of a sample being assigned to a treatment group and could be learned by a classification model (such as support vector machine used in this work). Therefore, 1) the augmented minibatch would contain the same number of samples for each treatment group; 2) different treatment group were balanced.

Third, a model selection strategy was proposed by estimating the PEHE using nearest neighbor search.

Comments:
This paper is well motivated. The key challenges in counterfactual inference is how to adjust for the bias in treatment assignment and the associated discrepancies in the distribution of different treatment groups.

The main idea of this paper, i.e., augmenting the minibatch through propensity score matching for each sample, is well explained in Section 3. However, it could be better if the introduction of model architecture (in Appendix F) was presented in the method section.

Did the author need to train (k choose 2) SVMs to compute the propensity scores for samples from k treatment groups?

When comparing different approaches, as were shown in Table 3, 4 and Figure 3,4, did the author run any statistical test, such as t-test, to confirm the difference between those distributions were significant? The standard deviations of those errors seemed quite large so the difference could be non-significant.

Could the author provide more explanations on why the proposed approach, i.e., minibatch augmentation using propensity score matching, can outperform the TARNET? In TARNET, each sample it only used to update the head network corresponding to the sample's treatment assignment, why would balancing samples in the minibatch can improve the estimation of treatment effect?

---

> ### Author Response · Authors · 2018-11-15
> **Rebuttal Reviewer 3**
>
>
> We thank Reviewer 3 for the helpful and constructive feedback. We address the feedback below in order.
>
> R3: "However, it could be better if the introduction of model architecture (in Appendix F) was presented in the method section."
>
> We thank R3 for this feedback. We have moved the architectural details to the method section in the revised draft.
>
> R3: "Did the author need to train (k choose 2) SVMs to compute the propensity scores for samples from k treatment groups?"
>
> No, only one multi-class SVM - one entry in the output vector for each treatment option - was necessary (see Algorithm 1).
>
> R3: "(...) did the author run any statistical test, such as t-test, to confirm the difference between those distributions were significant?"
>
> We have added the requested statistical significance test results (alpha < 0.05) to Tables 3 and 4. With few exceptions, most results on IHDP and News-2/4/8/16 were significantly different to PM due to the large number of repeated experiments. However, we found that on Jobs, where PM had the second best result after GANITE, and on News-4, where PM had the second best result after PD, those results were not significantly different.
>
> R3: "Could the author provide more explanations on why the proposed approach, i.e., minibatch augmentation using propensity score matching, can outperform the TARNET? In TARNET, each sample it only used to update the head network corresponding to the sample's treatment assignment, why would balancing samples in the minibatch can improve the estimation of treatment effect?"
>
> We thank R3 for this comment. In TARNET, the shared lower layers are trained on all samples (see p.4 last paragraph, or Shalit et al. 2017) and are therefore subject to the treatment assignment bias present in observational data. The TARNET architecture by itself therefore does not address treatment assignment bias. PM is orthogonal to TARNET, and improves upon the performance of TARNET by controlling for treatment assignment bias.

---

### Official Review · AnonReviewer1 · 2018-11-02
**Review: Interesting paper and impressive results; why novel vs. standard PSM?**

**Rating:** 5
**Confidence:** 3

**Review:**

========= Summary =========

The authors propose a novel method for counterfactual inference (i.e. individual/heterogeneous treatment effect, as well as average treatment effect) with neural networks. They perform propensity score matching within each minibatch in order to match the covariate distributions during training, which leads to a doubly robust model.

PM is evaluated on several standard semi-synthetic datasets (jobs, IHDP, TCGA) and PM shows state-of-the-art performance on some datasets, and overall looks quite promising.

======= Comments =======

The paper is well-written, presents a novel method of some interest to the community, and shows quite good performance across a range of relevant benchmarks.

I have one major issue with this work: I don't see why propensity-score matching *within* a minibatch should provide a substantial improvement over propensity-score matching across the dataset (Ho et al 2011). I find the cursory explanation given ("it ensures that every gradient step is done in a way that is approximately unbiased") unconvincing, since (a) proper SGD training should be robust to per-batch biases during training (the expected loss is identical for both methods, correct?), and (b) biases should go away in the limit of large batch sizes. If indeed SGD required unbiased *minibatches* then standard minibatch SGD wouldn't work at all.

Looking at the experimental details in the appendix, it appears that the MatchIt package was used to do PSM, rather than a careful comparison under the same conditions. Are the exact matching procedure, PS estimator model, choosing "one of 6 closest  matches by propensity score", batch size, etc. the same between your PM implementation and MatchIt? I'd be very curious to see the results of a controlled comparison between Alg S1 and S2 under the same conditions (i.e. run your PM implementation on the whole dataset), and perhaps even some more clever experiments illustrating why matching within a minibatch is important.

Another hypothesis for why PM is better than PSM is that the matching distribution for PM changes at each epoch (at least due to the randomization among the 6 closest matches). Could it be that the advantage of PM is that it actually provides a randomized rather than constant distribution of matched points?

Can the authors provide more motivation for why PM should outperform PSM? Or some more careful comparison of these methods isolating the benefits of PM? I think a convincing justification and comparison here could change my opinion, as I like the paper otherwise. Thanks!

Detailed Comments:

- There is insufficient explanation of the PM method in the main text. The method is only mentioned in a single sentence buried in the middle of a long paragraph "In PM, we match every sample within a minibatch...". This should be made more clear, e.g. by moving Algorithm S1 to the main text.
- The discussion on Model Selection and the argument for nearest-neighbor PEHE is clever and well-supported by the experiments.
- In Table 3 and 4, it's not clear which numbers are reported by the original authors and which were replicated by the authors.

---

> ### Author Response · Authors · 2018-11-15
> **Rebuttal Reviewer 1**
>
>
> We thank Reviewer 1 for the helpful and constructive feedback. We address the feedback below.
>
> R1: "Looking at the experimental details in the appendix, it appears that the MatchIt package was used to do PSM, rather than a careful comparison under the same conditions. (...) I'd be very curious to see the results of a controlled comparison between Alg S1 and S2 (...), and perhaps even some more clever experiments illustrating why matching within a minibatch is important. "
>
> We thank R1 for these insightful and important comments. In the latest revision, we have added the requested baseline that performs PSM using the PM matching procedure on the dataset level (called PSM_PM) . The old PSM baseline that used the "MatchIt" package is now referred to as PSM_MI. Although PSM_PM indeed performed better than PSM_MI, PM still consistently, with the exception of Jobs, outperformed PSM_PM (Tables 3 and 4). To further elucidate why this is the case, we added a new experiment for which we compared the training dynamics of PM, PSM_PM and PSM_MI (Fig. 6, and p.9 second to last paragraph).
>
> R1: "I have one major issue with this work: I don't see why propensity-score matching *within* a minibatch should provide a substantial improvement over propensity-score matching across the dataset (Ho et al 2011)."
> R1: "(a) proper SGD training should be robust to per-batch biases during training (the expected loss is identical for both methods, correct?)"
>
> R1 is correct in that the expected observed factual loss is identical for both methods. However, when learning counterfactual representations from observational data, we are interested in reducing the unobserved counterfactual error using the observed factual loss. In observational data, we do not have direct access to the counterfactual error, and a gradient step that improves the factual error may not necessarily improve the counterfactual error. In fact, in practice, we often see the opposite, undesired behavior of the factual error decreasing and the counterfactual error increasing as the network overfits to the properties of the treated group (see Fig. 6 - PSM_MI for an illustration of this behavior).
>
> As we have outlined in the manuscript, it is possible to implicitly optimise the unobserved counterfactual error using the observed factual samples if (1) we manage to break the dependence of treatment assignment on $X$, and (2) we observe all relevant variables. The results in Fig. 6 show that attempting to ensure that each individual batch conforms to (1) has a significant variance-reducing effect compared to sampling batches that may not necessarily conform to (1) as well from a pre-balanced dataset. This indicates that PM effectively minimises either the number or magnitude of undesired steps that improve the factual error but do not improve (or worsen) the counterfactual error.
>
> R1: "(b) biases should go away in the limit of large batch sizes."
>
> R1 is correct that the performance advantage of PM over PSM_PM would disappear in the limit case - when the batch size equals the number of samples in the training set. However, we are not aware of any prior works that have shown that either very large batch sizes or matching within batches may be necessary to effectively learn counterfactual representations with neural networks when using PSM with SGD, nor that ensuring that each individual batch is balanced has a significant variance-reducing effect. In addition, the use of minibatching may be a necessity when the entire dataset does not fit in memory.
>
> R1: "(...) Could it be that the advantage of PM is that it actually provides a randomized rather than constant distribution of matched points?"
>
> We thank R1 for this feedback. We added new experiments to the latest revision in order to determine the impact of the randomisation introduced by matching to one of the k nearest neighbours at random (Table S1). We found no significant differences in performance both between using randomisation and not using randomisation (k=1), and between variants of PM that used different choices of $k$ ranging from 3-15. We therefore conclude that randomisation is not an essential component of PM.
>
> We also thank R1 for the detailed comments and suggested improvements, which we have implemented in the revised manuscript.
>
> R1: "- There is insufficient explanation of the PM method in the main text. (...) This should be made more clear, e.g. by moving Algorithm S1 to the main text."
>
> We have moved Algorithm 1 to the main text in the revised manuscript.
>
> R1: "- In Table 3 and 4, it's not clear which numbers are reported by the original authors and which were replicated by the authors."
>
> We have added detailed documentation on the origins of the reported performance numbers in Appendix I.

---

> > ### Comment · AnonReviewer1 · 2018-11-22
> > **I'm still not convinced about propensity matching within a minibatch**
> >
> > "As we have outlined in the manuscript, it is possible to implicitly optimise the unobserved counterfactual error using the observed factual samples if (1) we manage to break the dependence of treatment assignment on $X$, and (2) we observe all relevant variables. The results in Fig. 6 show that attempting to ensure that each individual batch conforms to (1) has a significant variance-reducing effect compared to sampling batches that may not necessarily conform to (1) as well from a pre-balanced dataset. This indicates that PM effectively minimises either the number or magnitude of undesired steps that improve the factual error but do not improve (or worsen) the counterfactual error."
> >
> > I'm still confused by this reasoning, because it doesn't conform to anything that's commonly observed in the literature for training models with SGD. After you apply PSM, then it's already the case that "we manage to break the dependence of treatment assignment on $X$" across the dataset. Perhaps I'm still missing something, but if computing a regression value requires matching for each minibatch to "avoid bad gradient descent steps", shouldn't this be required even in a dataset with *no* selection bias? (i.e. you should need to balance treatments within each minibatch). Is there evidence of this?
> >
> > Taking it one step further, your claim seems to imply that when training a model on imagenet you should make sure you have an equal number of training examples from each class... ?

---

> > > ### Author Response · Authors · 2018-11-25
> > > **Rebuttal Reviewer 1**
> > >
> > > We thank R1 for the follow-up comment, and for extending to us the opportunity to clarify open questions.
> > >
> > > R1: "I'm still confused by this reasoning, because it doesn't conform to anything that's commonly observed in the literature (...)"
> > >
> > > We believe that our results are not at odds with prior literature on mini-batch SGD. The additional variance introduced by mini-batch SGD due to stochastic noise is well-reported [3], and several recent works show that the added variance of mini-batch SGD can be reduced by importance sampling strategies in the standard supervised setting [1,2,3]. Our work follows the same line of reasoning.
> > >
> > > In the standard supervised setting, the variance added by minibatch SGD leads to slower convergence [3], but will not typically lead to models that perform worse if they are trained until a convergence criterion has been met, e.g. early stopping. It is therefore perhaps surprising that we observe a difference not just in convergence speed (Fig. 6), but also in counterfactual estimation performance (Tables 3 and 4) when comparing models trained with and without batch matching.
> > >
> > > The cause of the observed difference in treatment effect estimation performance is that we do not have access to the counterfactual error in observational data to select the best model encountered during training. We therefore have to resort to using the factual error (e.g. factual MSE) or an estimator based on the factual error (e.g. NN_PEHE) to select the best model. However, when the variance during training is high, chances are high that we select a model based on the factual error that happens to be suboptimal. Through this mechanism, increased variance during training for counterfactual inference is directly linked to worse expected performance in treatment effect estimation. We do not observe the same behavior in standard supervised learning tasks because we have direct access to the underlying objective, and are therefore able to select the best encountered model regardless of the variance during optimisation, i.e. models that during training do not perform well will not be selected because their measured error is high.
> > >
> > > We added a Figure illustrating this mechanism graphically in Appendix D (Figure S1). We have since also looked at how the individual gradient steps behave during training (Figure S2), and found that the individual gradient steps for the counterfactual error in PSM_PM and PSM_MI are more frequently in the opposing direction of the factual error (off-diagonal entries in the heatmap) than when using PM.
> > >
> > > R1: "(...) shouldn't this be required even in a dataset with *no* selection bias? (i.e. you should need to balance treatments within each minibatch). Is there evidence of this?"
> > >
> > > Our experiments show that the relative difference in effect estimation performance between using batch matching (PM) and not using batch matching (TARNET) decreases when the treatment assignment bias (\kappa) is lower (Fig. 4). This is unsurprising because the factual error gradually becomes a better estimator of the counterfactual error as the treatment assignment bias disappears. In the limit case, when there is no treatment selection bias, the cause of the performance difference would therefore be removed entirely.
> > >
> > > R1: "(...) your claim seems to imply that when training a model on imagenet you should make sure you have an equal number of training examples from each class?"
> > >
> > > Since ImageNet is a supervised learning task, one is able to evaluate the true objective function, and can therefore directly select models by the observed factual error. We would therefore not expect a performance improvement from decreasing the variance during training.
> > >
> > > To summarise, in order to address the concerns raised by R1:
> > > - We have added the requested PSM_PM baseline, and shown that it consistently, with the exception of Jobs, performs worse than PM (Tables 3 and 4).
> > > - We have shown that this outperformance is not caused by the randomisation introduced by PM (Table S1).
> > > - We have added new figures to the manuscript that demonstrate that using PM improves the convergence and reduces the variance when learning counterfactual representations compared to using PSM_PM and PSM_MI.
> > > - In the latest revised draft, we have removed the original statement that was found to be confusing by R1, and given a clear connection between reduced variance during training and improved model performance in estimating treatment effects, caused by the inability to access the true counterfactual error in observational data (Appendix D).
> > >
> > > Let us know whether you believe your original questions have been addressed by the clarifications listed above.
> > >
> > > [1] Zhao et al. "Accelerating minibatch stochastic gradient descent using stratified sampling." arXiv:1405.3080 (2014).
> > > [2] Zhao et al. "Stochastic optimization with importance sampling for regularized loss minimization." ICML (2015).
> > > [3] Csiba et al. "Importance sampling for minibatches." JMLR (2018).

---

> > > > ### Comment · AnonReviewer1 · 2018-11-27
> > > > **Response to Rebuttal**
> > > >
> > > > Thank you for the detailed response; I appreciate the more detailed arguments about precisely why importance sampling within a minibatch (and reducing variance within a minibatch) is of particular importance for counterfactual inference. These would make a nice addition to the paper.
> > > >
> > > > Lets discuss the main point:
> > > >
> > > > "The cause of the observed difference in treatment effect estimation performance is that we do not have access to the counterfactual error in observational data to select the best model encountered during training. We therefore have to resort to using the factual error (e.g. factual MSE) or an estimator based on the factual error (e.g. NN_PEHE) to select the best model."
> > > >
> > > > If I understand correctly, you are saying that in a supervised task, if you don't use importance sampling then the MSE will bounce around a lot, and we only achieve good performance by selecting the model at exactly the correct minibatch where the objective is minimized (do you mean MSE on training set or validation set??) But in fact early stopping is typically performed by only selecting the *epoch* that minimizes validation error *on the entire validation set* (not the exact minibatch where some minibatch objective is minized).
> > > >
> > > > So perhaps you're claiming that the benefit of PM is that it allows the minibatch (factual) error to more correctly select that correct epoch to stop training. However, shouldn't early stopping be decided based on the epoch that minimizes the error on the *entire validation set* (rather than any single minibatch)? If PSM is used for the validation set, then the error on the validation set should be an unbiased estimator of the *counterfactual* error, so again I don't see the issue.
> > > >
> > > > Despite my objections, I do admit that this discussion has been quite interesting, and there is some value in this work in allowing a broader discussion of these questions.

---

> > > > > ### Author Response · Authors · 2018-11-27
> > > > > **Rebuttal Reviewer 1**
> > > > >
> > > > > We thank R1 for engaging in the discussion. We address the individual comments below.
> > > > >
> > > > > R1: "If I understand correctly, you are saying that in a supervised task, if you don't use importance sampling then the MSE will bounce around a lot, and we only achieve good performance by selecting the model at exactly the correct minibatch where the objective is minimized (do you mean MSE on training set or validation set??) But in fact early stopping is typically performed by only selecting the *epoch* that minimizes validation error *on the entire validation set* (not the exact minibatch where some minibatch objective is minized)."
> > > > >
> > > > > R1 is correct with the exception of one point: Throughout the manuscript, model selection and error computation were consistently performed on the level of epochs using the entire validation set, and our argument was indeed about model selection on the level of epochs. We did not use single minibatches to compute errors. We will add a clarifying statement to the caption of Fig. S1.
> > > > >
> > > > > Note that we have shown significantly increased variance when comparing PSM_PM and PM on both resolutions (per epoch and per batch):
> > > > > - Fig. 6 displays the variance difference between PM, PSM_PM, and PSM_MI across epochs measured using the entire validation set.
> > > > > - Fig. S2 shows the variance difference across individual gradient updates using one minibatch from the training set to compute the gradient update. The differences in error were again measured using the entire validation set.
> > > > >
> > > > > R1: "However, shouldn't early stopping be decided based on the epoch that minimizes the error on the *entire validation set* (rather than any single minibatch)?"
> > > > >
> > > > > R1 is correct that the entire validation set should be, and was, used to measure the factual error (see above).
> > > > >
> > > > > R1: "So perhaps you're claiming that the benefit of PM is that it allows the minibatch (factual) error to more correctly select that correct epoch to stop training."
> > > > >
> > > > > No, our claim was about the validation set error measured across epochs.
> > > > >
> > > > > R1: "If PSM is used for the validation set, then the error on the validation set should be an unbiased estimator of the *counterfactual* error, so again I don't see the issue."
> > > > >
> > > > > Matching, both on the batch level and the dataset level, is an approximate method for balancing covariates. In general, complete removal of the treatment assignment bias can not be expected in non-trivial datasets. For example, a commonly encountered issue with matching-based approaches is that there are not always matches of sufficiently high-quality available. The factual error measured on a pre-matched dataset is therefore in general not an unbiased estimator of the counterfactual error.
> > > > >
> > > > > The PSM_PM and PSM_MI subfigures in Fig. 6 show that the factual error retains a significant bias in estimating the counterfactual error on News-8 even after pre-matching using two different matching methodologies.
> > > > >
> > > > > In addition, if matching produced completely unbiased datasets, we would expect that PSM_MI and PSM_PM in Table 3 and 4 perform either better or the same as other methods (e.g. PM, + on X, CFRNET) that only differ in how they address treatment assignment bias but otherwise used the same model, training procedure and hyperparameter ranges. Instead, we found that all of these methods, PM, + on X and CFRNET, consistently, with the exception of Jobs, outperformed both PSM_MI and PSM_PM. Since said difference disappears when we remove the treatment assignment bias with all other factors held equal (Fig. 4), their outperformance is likely attributed to their ability to better address issues caused by the treatment assignment bias.
> > > > >
> > > > > Please let us know whether our comments have helped resolve your questions.

---

> > > > > > ### Comment · AnonReviewer1 · 2018-11-27
> > > > > > **Rebuttal Reviewer 1**
> > > > > >
> > > > > > Okay, let me try to paraphrase you once more:
> > > > > >
> > > > > > In counterfactual inference, you need to use error on the (propensity-score matched etc.) validation set (which you call 'factual error') as an early stopping criterion, however since matching procedures are not perfect, this proxy objective does not accurately estimate the true counterfactual error objective. Therefore, you are not necessarily able to select the correct early stopping time.
> > > > > >
> > > > > > Given that, you would like to avoid high variance in the model performance on the counterfactual objective so that the model you pick (i.e. the one that has low error on the matched validation set) doesn't happen to have high error on the counterfactual objective.

---

> > > > > > > ### Author Response · Authors · 2018-11-27
> > > > > > > **Rebuttal Reviewer 1**
> > > > > > >
> > > > > > > R1: "In counterfactual inference, you need to use error on the (propensity-score matched etc.) validation set (which you call 'factual error') as an early stopping criterion, however since matching procedures are not perfect, this proxy objective does not accurately estimate the true counterfactual error objective. Therefore, you are not necessarily able to select the correct early stopping time."
> > > > > > >
> > > > > > > Yes, that is correct. We typically do not have access to the true counterfactual error outside of synthetic datasets, and must therefore use the factual error (or a derived measure such as NN-PEHE) for model selection. This is typically not a perfect measure of the counterfactual error (Fig. 2).
> > > > > > >
> > > > > > > R1: "Given that, you would like to avoid high variance in the model performance on the counterfactual objective so that the model you pick (i.e. the one that has low error on the matched validation set) doesn't happen to have high error on the counterfactual objective."
> > > > > > >
> > > > > > > Yes, that is also correct. We believe that to be the primary advantage of using PM over PSM. However, there is a second condition: Next to low variance, we also desire the alignment of the optimisation of the factual and counterfactual error, i.e. during training the factual error should decrease when the counterfactual error decreases, ideally with the same relative magnitude (this is the "desired" state shown in Fig. S2). Even if the variance is low, results will also be suboptimal if the counterfactual error moves in the opposite direction of the factual error, as would be the case when a model is overfitting to the properties of the treated group (see e.g. Fig 6 PSM_MI). Fig. 6 and S2 show that PM provides significantly better alignment and lower variance than both PSM_MI and PSM_PM.

---

### Author Response · Authors · 2018-11-15
**Revised manuscript uploaded**

We thank the reviewers for their helpful and constructive feedback. We uploaded a revised draft to implement the suggested improvements and to address the concerns of the reviewers.

In addition, while performing the additional experiments requested by the reviewers, we discovered a small error in the code for the experiments for the "+ on X" and "+ MLP" ablations that could in some cases deactivate matching when it was supposed to be active. We have fixed the error and revised the reported numbers for the "+ on X" and "+ MLP" ablations in Tables 3 and 4. The "+ MLP" and "+ on X" ablations are now more competitive with PM on IHDP. The overall conclusions have not changed since both ablations still suffer from the curse of dimensionality in higher-dimensional datasets if not accounted for by, e.g., matching on a low-dimensional representation of X (see the rebuttal to R2; or p.8 paragraph "Counterfactual Inference.").  To ensure other results were not affected, we have also re-run all other experiments that used batch matching.

---

### Meta-Review · Area_Chair1 · 2018-12-15
**A well composed, novel contribution to counterfactual inference with neural nets but lingering questions remain about empirical significance.**

**Confidence:** 5
**Recommendation:** Reject

**Metareview:**

The reviewers found the paper to be well written, the work novel and they appreciated the breadth of the empirical evaluation.  However, they did not seem entirely convinced that the improvements over the baseline are statistically significant.  Reviewer 1 has lingering concerns about the experimental conditions and whether propensity-score matching within a minibatch would provide a substantial improvement over propensity-score matching across the dataset.  Overall the reviewers found this to be a good paper and noted that the discussion was illuminating and demonstrated the merits of this work and interest to the community.  However, no reviewers were prepared to champion the paper and thus it falls just below borderline for acceptance.